# REPRESENTING VALUE FUNCTIONS IN POWER SYSTEMS USING PARAMETRIC NETWORK SERIES

## ABSTRACT

We describe a novel architecture for modeling the cost-to-go function in approximate dynamic programming problems involving country-scale, real-life electrical power generation systems. Our particular scenario features a heterogeneous power grid including dozens of renewable energy plants as well as traditional ones; the corresponding state space is in the order of thousands of variables of different types and ranges. While Artificial Neural Networks are a natural choice for modeling such complex cost functions, their effective use hinges on exploiting the particular structure of the problem which, in this case, involves seasonal patterns at many different levels (day, week, year). Our proposed model consists of a series of neural networks whose parameters are themselves parametric functions of a time variable. The parameters of such functions are learned during training along with the network parameters themselves. The new method is shown to outperform the standard backward dynamic programming program currently in use, both in terms of the objective function (total cost of operation over a period) and computational cost. Last, but not least, the resulting model is readily interpretable in terms of the parameters of the learned functions, which capture general trends of the problem, providing useful insight for future improvements.

Keywords: energy dispatch, approximate dynamic programming, cost function approximation, artificial neural networks, parametric network series, reinforcement learning.

## 1 INTRODUCTION

The operation of electrical energy systems involving a mixture of thermal and renewable sources is particularly challenging due to a number of factors. The value of water stored in the dams needs to be assessed constantly depending on the availability of other sources and short, middle and long term forecasts, which are highly variable and depend on the seasons in complex ways. Other renewable sources such as wind and solar power also depend on highly variable factors. Thermal power plants, on the other hand, need to be bootstrapped over relatively long periods before reaching full capacity, so that the effect of turning them on and off also has lasting implications. Last but not least, the demand of electricity is highly variable and exhibits complex seasonal patterns at different scales.

The concrete case that we are dealing with is the country-wide energy dispatch of Uruguay. This paper presents the preliminary implementation of a novel method for computing the optimal operation method of the Uruguayan grid. The method has been implemented within the Electric Energy Systems Platform (SimSEE).[1] The SimSEE system is currently in use by the Electric Market Administration (ADME) of Uruguay for continuously programming the optimal operation of the Uruguayan electrical system at several different time scales. Besides Uruguay, the SimSEE system is currently in use in República Dominicana and Belize.

Besides optimizing the schedule, SimSEE is also capable of producing detailed simulations (through precise physical models) of the evolution of the system in terms of the actual continuous state-space. This feature is key in the development of our proposed method, which can be seen as a plug-in replacement for the operation optimization component of SimSEE.

---

[1]{SimSEE}https://simsee.org

Methods based on Approximate Dynamic Programming (ADP) (Bertsekas & Tsitsiklis, 1996; Sutton, 1995), are the usual choice for optimizing the operation of energy systems. These methods rely on estimating the cost of operation for any possible initial state of the system. Unfortunately, when the number of state variables is large, ADP methods suffer from the so called Bellman's curse of dimensionality (Bellman, 1957), meaning that the number of states and actions on which the cost function needs to be evaluated grows prohibitively large and thus cannot be reliably estimated. This is true even for sophisticated variants of ADP such as Stochastic Dual Dynamic Programming (SDDP) (Pereira & Pinto, 1991). Moreover, the SDDP method is particularly sensitive to inputs with high variability such as renewable-energy sources.

The aforementioned problem arises when attempting to explicitly evaluate all possible values of the state variable, whose number grows exponentially with the dimension of the state space. This is obvious even if the state variables are discrete. When the variables are continuous, traditional ADP methods transform them into discrete variables via some form of quantization. In such scenarios, a common strategy is to employ dimensionality reduction techniques such as Principal Component Analysis (PCA) (Jolliffe, 2005). Of course, the price of quantization and dimensionality reduction is that the true value function can only be evaluated in an approximate way.

The current method used by SimSEE is a traditional Backward-ADP recursion applied over a quantized, reduced state-space of the whole Uruguayan system. The recent diversification of the power generation matrix has exerted a significant stress on the aforementioned method. This is particularly so for the short term operation, which is re-computed hourly.

## 1.1 VALUE FUNCTION APPROXIMATION

A recent alternative approach to the above techniques is to construct an implicit representation of the value function (Powell, 2011; Sutton & Barto, 2018). Instead of reducing the dimensions and/or quantizing the state space, a continuous model is built in the original state space based on a set of values of the function evaluated at arbitrary positions. Here, a rich set of tools from Approximation Theory is available to choose, construct and evaluate the appropriate approximation method for a particular task. Kernel methods (Xu et al., 2014) are a popular choice which is backed by the elegant theory of Reproducing Kernel Hilbert Spaces (Paulsen & Raghupathi, 2016). Another natural family of methods, more flexible, but also harder to characterize, is that of artificial neural networks (ANN), which are well known for being universal function approximators (R.Barron, 1994).

The above methods, however, have their drawbacks too. For instance, it has been shown that, if no additional measures are taken, the number of samples of the value function required for the overall approximation-based ADP method to converge can be even larger than that required by using traditional, explicit evaluation methods (Du et al., 2020). Luckily, such requirement can be significantly relaxed if appropriate variance reduction techniques are applied, e.g., *Common Random Numbers* (Christophe et al., 2015).

In the particular case of ANN approximators applied to an heterogeneous energy system, the above measure might not be enough. The great flexibility of ANNs also implies a great sensitivity to the input data, which in our case is highly variable due to the random nature of renewable energy sources. In this challenging scenario, further measures need to be taken in order to obtain parsimonious approximations.

## 1.2 PARAMETRIC NETWORK SERIES

Luckily for us, the signals and processes involved in the planning of energy dispatch usually exhibit regular patterns. This can be exploited to impose parsimonious approximations which extrapolate reasonably to unseen states. Our proposed method combines the flexibility of ANNs with prior information about the problem. In a nutshell, the value function, which is a function of state and time, is approximated by one neural network per time slot. The architecture of the network is the same for all time slots, reflecting the fact that the structure of the system itself does not change. The parameters do vary across networks, albeit in a controlled fashion: for any given link in the architecture, the corresponding weight is a function of time. The general idea is depicted in Figure 1.

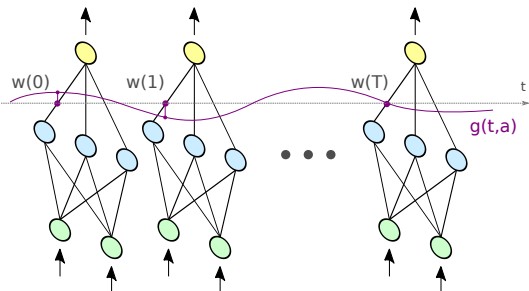

Figure 1: Parametric network series. When re-computing the hourly model, we can do a warm-restart by droping the leftmost network and adding a new one to the right.

### 1.3 WHY NOT A SINGLE NET?

Naturally, one could use a single parametric function to model the whole value function across all time steps. The reason for us to opt for a sequence of smaller models is again computational performance: instead of computing the whole approximation from scratch at each time step (usually an hour), we can quickly update the overall model by dropping the first model (at $t = 1$), adding a new one at the end, and running a few iterations. We call this a *sliding window* strategy, in reference to similar modeling patterns used in other fields.

### 1.4 RELATED WORKS

As mentioned, the use of approximation in value functions is an established technique described in classic textbooks such as (Bertsekas & Tsitsiklis, 1996; Powell, 2011; Sutton & Barto, 2018). Also, incorporating time dependence in approximation models and, in particular, artificial neural networks, is an active line of research. The main difference between the related works in this matter is in *how* the time dependence is imbued into the architecture.

Recurrent Neural Networks (RNNs) are widely deployed to capture time-dependent patterns using their current widely adopted incarnation, the so called Long-Short Term Memory (LSTM) networks (Hochreiter & Schmidhuber, 1997). In these models, the hidden layer outputs depend not only on their current input, but also on their own previous output, in a way similar to a recursive filter. Such architectures are able to produce outputs which depend not only on the current input, but on past inputs as well. However, these kind of architectures are incapable of capturing periodic patterns, especially if such patterns have low frequencies, such as periodic seasonal patterns.

Another widespread technique to represent time in ANNs is generally known as *positional encoding* (see Zheng et al. (2021) for a recent review on the subject). The typical positional encoding method involves auxiliary sinusoidal inputs to the network, usually several of them, with different frequencies. Although their typical use is in the context of language translation (see e.g. (Gehring et al., 2017)), where the frequencies are high (letter, word, phrase, a few time steps), nothing prevents one from using such techniques in a scenario such as the one described in this paper.

The main difference between the "positional encoding" approach and our proposed method lies in the way that this periodical information is fed into the system. Whereas the former uses these auxiliary signals as inputs to the whole network, and their influence on the weights is implicitly learned through standard backpropagation techniques, our method imposes a periodic pattern explicitly on the weights themselves, giving them some room to accomodate for the particularities not captured by the periodic model.

### 1.5 CONTRIBUTIONS

In summary, the main contributions of this work are two. First, we develop a novel architecture for learning value functions in approximate dynamic programming problems; the novelty lies in the use of a series of neural networks of identical architecture, one per time slot, where the weights are functions of time rather whose parameters are learned at training time. Second, we demonstrate the use

of the proposed model on a real-life, complex case, where we can evaluate the actual performance of the model in terms of real operation cost savings.

### 1.6 DOCUMENT ORGANIZATION

The rest of the document is organized as follows. In Section 2 we provide a formal introduction to the problem and the notation used throughout the paper. Section 3 introduces the problem of electric systems operation. The proposed method is described in detail in Section 4.4. Section 5 shows a detailed comparison between the current model in production and the proposed method, and discusses the results. Concluding remarks are given in Section 6.

## 2 PRELIMINARIES

We follow the usual notation and conventions used in approximate dynamic programming. The task is to operate a system over a time period in a way that minimizes the total operation cost over that period. The time is observed and operated at regular time intervals $tT$. In energy systems, $T$ may represent an hour, a day, a week, etc. Here $t$ is the discrete time index. The state of the system at time $t$, represented by $x_t \in \mathbb{X}$, evolves according to the following rule:

$$x_{t+1} = f(x_t, u_t, w_t, t).$$

The vector $u_t \in \Omega_t$ represents the controllable variables (the action), $w_t$ is a vector of exogenous, random, uncontrollable variables, and $f(\cdot)$ is the *state evolution function*. Note that, as the notation implies, the feasible set of actions $\Omega_t$ may vary through time. In the case of electric power systems, the function $f(\cdot)$ can be modeled with great precision and thus it can be assumed known. Then, it is possible to compute, the cost, at time $t$, of taking a particular action $u_t$ while in state $x_t$, given the exogenous variables $w_t$. We call this the *stage cost function* and write it as $c(x_t, u_t, w_t, t)$.

### 2.1 THE FUTURE COST-TO-GO, OR VALUE, COST FUNCTION

When we are about to take a given action, it is not the stage cost that we are interested in, but its long term impact in the future cost of the system. Unfortunately, the system is subject to random fluctuations due to $w_t$, making the future cost a random variable. The future cost function $J(x_t, t)$ is the expected value of the future cost, assuming that one can take the best action for every possible realization of $w_t$. The value of $J(x_t, t)$, in turn, can be decomposed as the stage cost at time $t$, and the future cost at time $t + 1$ and state $x_{t+1}$, leading to the well known Bellman recursion:

$$J(x_t, t) = \mathbb{E}_{W_t} \left[ c(x_t, u_t, w_t, t) + q \cdot J(f(x_t, u_t, w_t, t), t + 1) \right], \tag{1}$$

where $u_t$ is the best possible action given the current state $x_t$ and a particular realization of the random inputs $w_t$, which is represented by the random variable $W_t$, and $q \in (0, 1]$ is a *discount* factor, generally defined by the compound interest rate of the currency (in this case, USD):

$$u_t = \arg \min_{\zeta \in \Omega_t} \{ c(x_t, \zeta, w_t, t) + q \cdot J(f(x_t, \zeta, w_t, t), t + 1) \}. \tag{2}$$

### 2.2 VALUE ITERATION

The above equation defines a mapping between the current state $x_t$, a realization of $w_t$, and the action $u_t$. We call this mapping an *operation policy*. Ideally, if $J(x_t, t)$ was perfectly known to us, the policy obtained above would be the optimal policy of the system. However, there is a clear "chicken-and-egg" problem between $u_t$ and $J(x_t, t)$, and so practical solutions need to provide a starting point.

One of the well established methods in the theory of approximate dynamic programming (see (Bertsekas & Tsitsiklis, 1996; Powell, 2011; Sutton & Barto, 2018)) is to transform the above recursive problem into an iterative one, by starting with an arbitrary future function approximation $\tilde{J}^{(0)}(x_t, t)$ (here the superindex $(0)$ refers to the initial iteration) and then estimating $\tilde{J}^{(k)}(x_t, t)$ for $k = 1, \ldots$ until some convergence criterion is met.

Table 1: STM state space description.

| variable | Description | cardinality |
|---|---|---|
| 1 | Volume in lake Rincón de Bonete | 10 |
| 2 | Combined cycle boiler #1 blowdown (4h) and steam turbine #1 loading (2h) | 6 |
| 3 | Combined cycle boiler #2 blowdown (4h) and steam turbine #2 loading (2h) | 6 |
| 4 | Combined cycle boiler #1 cooling time (120 h) | 3 |
| 5 | Combined cycle boiler #2 cooling time (120 h) | 3 |
| 6 | Volume in lake Palmar | 5 |
| 7 | Volume in lake Salto Grande | 5 |
| 8 | Runoff status of the Río Negro basin | 4 |
| 9 | Runoff status of the Uruguay River basin | 4 |

## 3 SCENARIO

### 3.1 BACKGROUND ON ELECTRICAL POWER SYSTEMS

Traditional, thermal power generating plants (natural gas, combined cycles, coal-fired plants, nuclear) have a high variable cost (measured in USD per MWh), but are very predictable. On the contrary, renewable sources such as hydroelectric, solar or wind have zero variable cost, but are highly unpredictable.

When renewable resources are abundant (sun, wind, etc.), the cost of supplying the demand may be $0$. On the other hand, when such resources are short, the system must rely on the highly costly thermal sources. Thus, the ever increasing addition of wind and solar energy in electrical power generation systems implies savings, but also greatly complicates planning, as the fluctuations in the stage costs become larger and more unpredictable. It is therefore very important to construct the best possible stochastic models for the resources involved. These models must capture the correlations between the different resources (e.g., wind at nearby solar plants) as well as their temporal dependencies. This subject is in itself a large area of active research. The interested reader is referred to (Flieller & Chaer, 2020).

The state transition function is also of paramount importance. Naturally, these functions are specific to the physical characteristics of each energy power generation system. For the particular case of the Uruguayan system see (Chaer & Monzon, 2008).

Last, but not least, the value function is modeled so that it measures the economic impact of the different actions. Different time scales involve different actions, each with its own range of options, and thus different value functions and operational restrictions are applied in each case. We provide more information on this in appendixes B and C.

Traditional value function approximation methods either construct independent estimations $\tilde{J}(x_t, t)$ for each time $t$, or attempt to construct a single $\tilde{J}(x_t, t)$ that works for all $t$. Our proposed method, which lies in between these two approaches, exploits the continuity in time of the value function to construct a series of approximations $(\tilde{J}(x_t, t), t = 1, \ldots, n_t)$ where the corresponding parameters $(\Theta_t, t = 1, \ldots, n_t)$ are themselves a function of the time $t$. The parameters of these guiding functions are themselves learned within the iterative approximation loop. The technical details of this method are described in the next section.

### 3.2 SHORT TERM (STM) OPERATION OF THE URUGUAYAN SYSTEM

In recent years, the Uruguayan system has undergone radical changes in its generation matrix (Cornalino et al., 2018) with the widespread incorporation of wind and solar sources. These changes have exerted considerable pressure on the current optimization tools in use, thus motivating the development of newer approaches such as the one presented in this work.

The STM contains 9 *visible* state variables (several other state variable exist, but are not shown since the optimal operation does not depend on them). Table 1 shows a detail of these variables; the last column shows the number of discrete values into which each variable is quantized when solving the problem using the classic Bellman recursion-based optimization method currently in production.

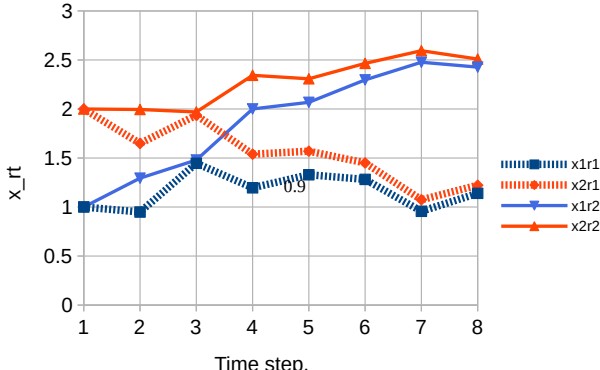

Figure 2: Depiction of the common random numbers in a one-dimensional state space. The blue and red trajectories correspond to two different initial states. The dotted trajectories correspond to one random seed, and the solid ones to another.

## 4 PROPOSED METHOD

The overall algorithm begins with an initial *continuous* value function estimation $\tilde{J}^{(0)}(x, t), x \in \mathbb{X}, t \in [n_t]$. Here $[n_t]$ is an abbreviation for $\{1, 2, 3, \ldots, n_t\}$, and $\mathbb{X}$ is the continuous state space. The superscript 0 indicates the initial iteration $k = 0$. Each iteration $k > 0$ involves three steps: i) simulation, ii) sampling of the value function (at discrete points), and iii) approximation (of the continuous value function) based on the aforementioned points. The following subsections describe each of these steps in detail, albeit for a generic architecture. The last subsection describes the specific architecture used in our case. A diagram of the overall algorithm is also provided in Appendix A.

### 4.1 SIMULATION USING COMMON RANDOM NUMBERS

In order to construct an approximation $\tilde{J}(x, t)$, the function $J(x, t)$ needs to be sampled at a number of points in state space. In order to obtain a set of interesting points on which to evaluate the function, as well as their corresponding values, we simulate the operation of the system subject to a number of different realizations of the external random variables (e.g., rain, sunshine, wind).

Starting at state $x_0 = x$, the system is allowed to evolve up to a pre-specified time horizon $n_t$ by drawing samples from $w_t \approx W_t$, computing the optimal action $u_t$, and updating $x_{t+1} \leftarrow f(x_t, u_t, w_t, t)$. A well known issue with this scheme is that the variance of the estimated variable $J(x, t)$ grows quickly as the horizon $n_t$ increases, thus limiting the ability of the overall method to evaluate the long term impact of the chosen actions.

The *common random numbers* technique, depicted in Figure 2, greatly reduces the variance of the estimates (at the cost of increasing their bias) by using, for every simulation stage, a fixed set of $n_r$ pseudo-random ensembles $(\mathbf{w}_1, \mathbf{w}_2, \ldots, \mathbf{w}_{n_r})$. A given ensemble $\mathbf{w}_r = (w_{r0}, w_{r1}, \ldots, w_{rn_t})$ is called a *chronicle*.

Since we always begin from the same known state $x_0$, the simulation results in $n_r$ trajectories, one per chronicle. Since the trajectories depend globally on the current approximation of the value function, $\tilde{J}^{(k-1)}(x, t)$, a different set of trajectories will be obtained at each iteration $k$.

### 4.2 POINT-WISE EVALUATION OF THE VALUE FUNCTION

Once the $n_r$ trajectories are computed, the sample values $J^{(k)}(x, t)$ need to be estimated at all the states visited by all the trajectories. For this, we use a variant of the method described in (Sutton & Barto, 2018, Chapter 12) as *compound TD estimation*. Given a trajectory $\mathbf{x}^r = \{x_{rt}^{(k)} : t \in [n_t]\}$,

we compute the corresponding sample values $\{J^{(k)}(x_{rt}^{(k)}, t) : t \in [n_t]\}$ as follows:

$$\bar{J}^{(k)}(x_{rt}^{(k)}, t) = \sum_{\delta=0}^{\Delta} \mathrm{TD}_{\delta}^{(k)}(x_{rt}^{(k)}), \tag{3}$$

where

$$\mathrm{TD}_{\delta}^{(k)}(x_{rt}^{(k)}) = \sum_{d=0}^{\delta} q^d c(x_{r(t+d)}^{(k)}, u_{t+d}, w_{t+d}, t+d) \; + \; q^\delta \tilde{J}^{(k-1)}(x_{r(t+d)}^{(k)}, t+d).$$

In order to further enhance the overall stability of the method, each of the above estimates $\bar{J}^{(k)}(x_{rt}^{(k)}, t)$ is mixed with the value obtained using the current *continuous* approximation of $\tilde{J}^{(k)}(x_{rt}^{(k)}, t)$ (note that, in general, we will not have point-wise estimations of these values, as the trajectories from one iteration to the other are different). The mixture is computed as:

$$\hat{J}^{(k)}(x_{rt}^{(k)}, t) \leftarrow (1-\alpha)\tilde{J}^{(k)}(x_{rt}^{(k)}, t) + \alpha\bar{J}^{(k)}(x_{rt}^{(k)}, t), \tag{4}$$

where $0 < \alpha < 1$ is a parameter

### 4.3 VALUE FUNCTION APPROXIMATION

Note: for the rest of this section, we will omit the iteration index $k$ to clarify notation, as all related variables belong to the current iteration.

After simulation, we have $n_r$ state-value training pairs available to approximate the value function at each time slot $t > 0$. We will now describe the generic training procedure, leaving the details of the architecture for the next subsection.

The overall approximation is given by $n_t$ parametric models, one for each $t \in [n_t]$, and each of these models is adapted using the value function sampled at all states visited at time $t$ during the simulation stage. We call this set of points, $\{x_{rt} : r \in [n_r]\}$, a *constellation*. The training data at time $t$ is the set of pairs $(x_{rt}, \hat{J}(x_{rt}, t))$.

Given the training data, the approximation of the value function, and the time step $t$, $\tilde{J}(x, t), x \in \mathbb{X}$, depends on a set of $p$ parameters $\Theta_t = (\theta_{1t}, \theta_{n_p t})$; we write this dependency explicitly as $\tilde{J}(x, t|\Theta_t)$. As usual, these parameters are adapted so that the approximation error is minimized. This error is given by:

$$\sum_{r=1}^{n_r} \left[ \tilde{J}(x_{rt}, t|\Theta_t) - \hat{J}(x_{rt}, t) \right]^2. \tag{5}$$

Besides minimizing (5), each of the $n_p$ parameters of the approximation model is encouraged to follow a particular curve, which is itself parametric. Concretely, the $n_t$ values of the $p$-th parameter of the architecture across time, $(\theta_{p1}, \theta_{p2}, \ldots, \theta_{pn_t})$, are modeled after a function $g(t, \mathbf{a}_p)$,

$$\theta_{pt} \approx g(t, \mathbf{a}_p),$$

where $\mathrm{a}_p$ is the set of $n_a$ hyper-parameters associated to parameter $p$ which are themselves adaptive. Correspondingly, an additional term is added to the cost function to enforce that relation:

$$\frac{1}{2} \sum_{p=1, t=1}^{n_p, n_t} \beta_{pt} \left( \theta_{pt} - g(t, \mathbf{a}_p) \right)^2. \tag{6}$$

Using different penalties $\beta_{pt}$ we can modulate the degree of fitness to the guiding functions $g(t, \mathbf{a}_p)$ depending on the time index $t$ and/or the corresponding architecture parameter $p$. For example, we can ignore the imposition of a temporal pattern by setting the corresponding $\beta_{tp} = 0$ for a particular $p$ and all $t$.

Finally, the hyper-parameters $\mathbf{a}_p, p = 1, \ldots, n_p$ are regularized using an $\ell_2$ (Ridge) penalty with parameter $\mu$. The overall cost function to be minimized during learning is:

$$L(\Theta_t, \mathbf{a}_p : t \in [n_t], p \in [n_p]) = \frac{1}{2} \sum_{r,t} \left[ \tilde{J}(x_{rt}, t|\Theta_t) - \hat{J}(x_{rt}, t) \right]^2 \tag{7}$$

$$+ \frac{1}{2} \sum_{p,t} \beta_{pt} \left[ \theta_{pt} - g(t, \mathbf{a}_p) \right]^2 + \frac{\mu}{2} \sum_{p,h} a_{ph}^2$$

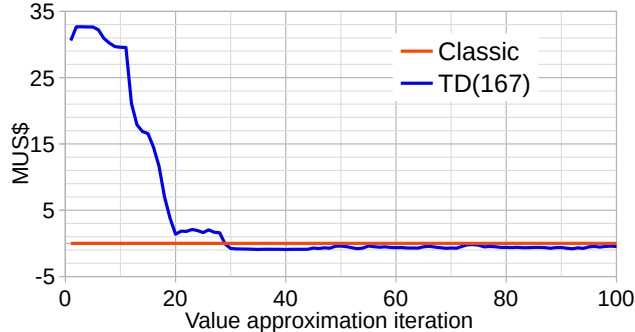

Figure 3: Difference between PDS for $\Delta = 167$ and the Classic algorithm as a function of the iteration. The PDS-based method surpasses the classic one after 30 iterations.

### 4.4 ARCHITECTURE

As mentioned in Section 3, electrical power systems exhibit extremely variable costs. Because of this, and despite the imposed regularity, the best architecture found to represent the value function at a given state for this (simplified) STM model consists a single hidden layer of 4 neurons with tanh activations, and a linear output. Recall now that we have an ensemble of $n_t = 240$ such networks, one per time step. The parameters of each of these networks are regularized by encouraging them to follow a smooth time-dependent function, as described before. In the case of the STM, we choose a third order polynomial:

$$g(t, a_{1p}, a_{2p}, a_{3p}, a_{4p}) = a_{1p} + a_{2p}t + a_{3p}t^2 + a_{4p}t^3.$$

For medium and long-term models, which are currently in development, we can use sinusoidal functions matching the seasonal patterns of both energy production and demand (day, week, year).

## 5 RESULTS

In this section we compare the total operation cost obtained with the new method and with the "Classic" Bellman iteration-based method currently in use. We evaluate the STM hourly models of both methods along the 10 day (240 hour) period that goes from 8/24/2021 9:00AM to 9/3/2021 9:00 AM.

We also set $\mu = 10^{-12}$ in (7), the discount factor $q = 1$, the number of chronicles used in the simulation to $n_r = 100$, $\Delta = 167$ in (3) and $\alpha = 0.3$ in (4).

Figure 3 compares the expected 10 day cost-to-go value obtained with the proposed and the one obtained using the Classic one. The new PDS-based method is seen to surpass the performance of the classic method after 30 iterations (more details on the evaluation of these policies are given in Appendix F).

Figure 4 shows the evolution of the cost-to-go derivatives with respect to the state variables for a given fixed state during the simulation horizon. As can be seen, the parsimony imposed on the parameters of the set of networks forces an evolution with similar parsimony in the derivatives. From the point of view of the operation of the system, parsimony in these derivatives is essential since these derivatives are transformed into operating instructions and it would not be feasible to abruptly change the dispatch of the generation units.

In terms of execution time, 30 iterations using the PDS approximation requires just 12 minutes to complete, whereas the classic Bellman recursion requires 31 *hours* to reach the same performance over the same STM state space (see Appendix E for details on the complexity of the CBS method for this scenario). It is important to remark that the optimal programming needs to be recomputed on an hourly basis, thus rendering the Classic approach infeasible for this scenario (the Classic method

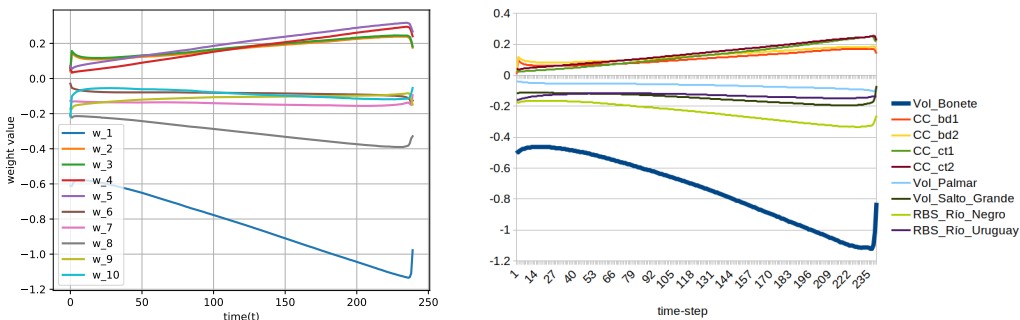

Figure 4: Here we show the effect of $g(t, a)$ on a few network weights (left), and the derivative of the cost-to-go function w.r.t the state variables (right), reflecting the parsimonious nature of the weights.

here was trained off-line). Thus, the new PDS-based method not only reduces times dramatically, but also allows us to obtain a better operation by considering a more complex state space.

## 6 CONCLUDING REMARKS

We have presented a new parametric value function approximation model capable computing the optimum programming of a large and diverse electrical power systems. We have tested this new method on the short term model of the real Uruguayan system (the most challenging one), and confirmed it to be superior to the one in production using drastically less resources. Also, proposed parametric network series (PNS) provides both stable and interpretable models by capturing the time-dependent trends in the network parameters.

We are currently working on implementing the *sliding window* strategy for obtaining even faster updates, as described in Section 1, and on deploying the new model on medium (MTM) and long term (LTM) scenarios.

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

## A  VALUE ITERATION LEARNING LOOP DIAGRAM

A diagram-box of overall algorithm is shown in Figure 5.

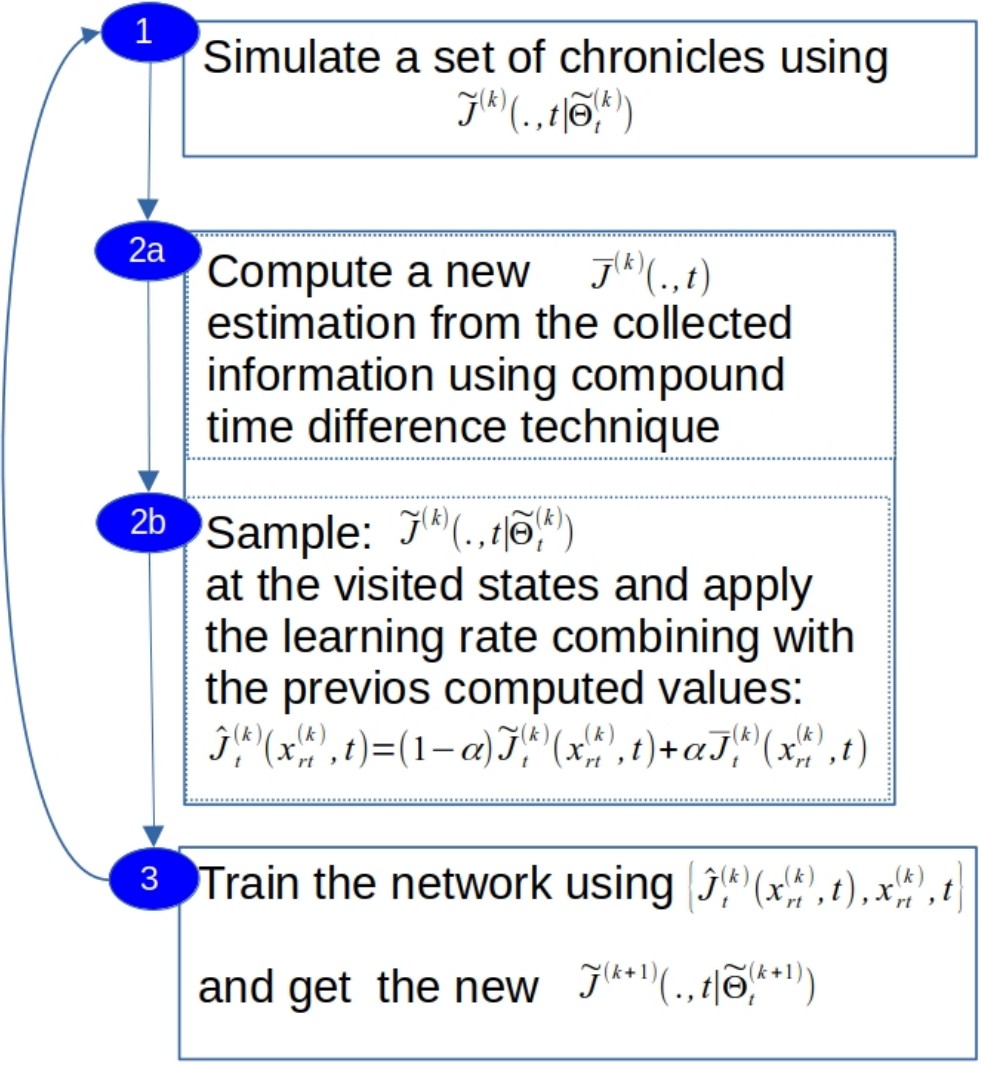

Figure 5: Value Iteration - Learning loop.

## B  VALUE FUNCTION IN ENERGY DISPATCH PROGRAMMING

In the generation system we have several time horizons to solve. The value function weights different operation costs depending on the horizons. In the medium term (several months) the value function assigns value to the water stored in the lakes with the highest capacity. This is used to trigger purchases of ships with fuel, which must be made 90 days in advance so that the fuel is available when required.

On a weekly basis, the operating policy takes into consideration the information of the rain, wind, solar forecasts, and the demand. As a result, the distributions of the exportable energy blocks (system surplus) and their recovery (minimum price to receive) are available one week in advance.

For daily simulations, the value function captures the effect of the forecasts of availability of water, solar and wind resources and allows determining the order of dispatch of the resources, the instructions given to the operators regarding the recovery of the dammed water in each of the lakes (derived from the value function with respect to using a unit of energy stored in each lake). This valuation allows the operators, in real time, to make decisions to maintain the power balance of the system while maintaining a dispatch of minimum cost. The same short-term simulations with hourly detail, allow to visualize the probabilities of dispatching thermal power plants and to give advance notice to the operators of said units so that they are ready to dispatch when requested.

## C  PENALTIES ON THE VIOLATION OF RESTRICTIONS

In the representation of the system used, if a restriction is included through a penalty, the value of the penalty is calculated by evaluating the real effect on the system when said restriction is violated. Therefore, the value function includes both the costs associated with the use of fuels and imports as well as the implicit costs associated with the penalties on the represented restrictions. As an example, it is a common practice to represent the power balance constraint at each node of the system as the sum of the generator powers minus the sum of the demands equal to zero, but this constraint is represented with penalties whose values are estimated as the cost of the country's economy for not supplying the demand.

## D  THE NEED TO RE-TRAIN THE MODEL AT EACH TIME STEP

Modern electric systems integrate several wind and solar energy sources. These sources have important fluctuations in terms of hours. It is the introduction of these short-term variabilities that makes the filtering elements (reservoirs, battery banks, etc.) of the system that were not important before, now do. The need to operate these new filtering elements, using the information from the forecasts, leads to an increase in the dimension of the state of the system to be considered. This change has been accentuated in the last 5 years and will continue in the same direction in the coming years as a worldwide trend. Theoretically, if the representation of the state of the system was complete, it would not be necessary to fit a model for each time step. But the representation of the state is never complete. It should be remembered that the structure of the system is variable over time. In the short term, the forecast information is represented based on time series of biases and attenuators that shift the distributions that model the associated stochastic processes. This leads to the same state (represented) of the system, but in different time steps, the value function is different. In the long term, the structure of the system changes as the demand grows, the power plants age and are withdrawn and new generators enter. On an annual scale, the different stations impose shifts in the distributions associated with wind, solar and hydraulic energy and on the Demand of the system.

## E  COMPUTATIONAL COST OF THE CLASSIC BELLMAN RECURSION

Table 1 shows the nine state variables considered in the model along with the corresponding number of discretized values. Solving a single time step in Classic Bellman's Recursion (CBR) requires one to compute the value function for every combination of the discrete values of all nine variables. In our case, there are $10 \times 6 \times 6 \times 3 \times 3 \times 5 \times 5 \times 4 \times 4 = 1 : 296000$ different states. This, in turn, has to be carried out for each time step. In our case we have 240 such stems, for a total of

311040000 value function estimations. On the other hand, in the proposed method, simulations of 100 chronicles are being carried out in each simulation cycle. Each chronicle implies computing the value at just the 240 visited states and therefore each simulation cycle implies solving the dispatch in $100 \times 240 = 24000$ states. The computational effort of the classical Bellman resolution is then $311040000/24000 = 12960$ times greater than that of an information collection cycle of the proposed algorithm. For the example presented, the proposed algorithm converged to a slightly better Operation Policy than the classical one in just 30 iterations of the learning loop.

## F   DETAILS IN THE EVALUATION OF THE POLICIES

First 100 iterations of the learning loop were performed (see Figure 5 with block diagram of the algorithm). Then, each of these iterations, the functions of value $\tilde{J}^{(k)}(.,t)$ were taken and the operation of the system with these functions was simulated for a set of 1000 new realizations of the stochastic processes (generated with new random seeds not used for training previous). Finally with these simulations, the estimate of the expected value of the future operation with each $\tilde{J}^{(k)}(.,t)$ was obtained. And with the same random seeds, it was simulated using the value function obtained with the classical Bellman algorithm (currently in use for dispatch resolution). The obtained estimations are shown in Figure 3

## G   ON THE CHOICE OF THE PARSIMONY CRITERION

There are two aspects for which the imposition of temporal parsimony seems important to us. The first is that in the actual operation there is a certain temporary behavior that must be respected. They are restrictions imposed by the operation. As an example, it would not be permissible to have operating instructions that imply dispatching a combined cycle power plant at one hour and shutting it down the next. These operating instructions arise directly from the derivatives of the value function with respect to the state variables and therefore, by giving temporary parsimony to the network parameters, indirectly the same type of parsimony is being imposed on the operation instructions.

The other aspect, no less important, is that it is a way of introducing expert knowledge, such as the trends and seasonality to be expected in the variables of the system (demand growth, rainy seasons, windy seasons, etc.). It would be possible to let the network learn these relationships from simulations, but remember that we are fighting the Bellman curse and that we intend to focus the neural network's representational capacity and computational effort efficiently. The implementation carried out allows to experience the effect of considering or not the temporal parsimony increasing or reducing the value of the penalties $\beta_{pt}$.

