# OpenReview forum: "Representing value functions in power systems using parametric network series"
_ICLR.cc/2022/Conference — ICLR 2022 Submitted_

### Official Review · Reviewer_wkjh · 2021-10-31

**Correctness:** 3
**Technical Novelty And Significance:** 1
**Empirical Novelty And Significance:** 2
**Recommendation:** 3
**Confidence:** 4

**Details Of Ethics Concerns:**

No.

**Main Review:**

## Strengths
1. The background knowledge is well written.
2. The presentation of the main contribution is clear enough.
3. A good attempt on combining data-driven ADP on the real-world problem.
4. Although the authors seem not a researcher in the ML community, the fundamental knowledge of ML described in the paper is almost correct and the logic is very clear.

## Weaknesses
1. The proposed method on representing the value function is actually not novel enough in the community of ML. It can be seen as a special case of using the hyper-networks to model the value function [1]. Nevertheless, the attempt to create innovations in the traditional fields should be encouraged.
2. Although the motivation of proposing time-dependent value function is correct, the representation and resolutions of **time steps** should be discussed. The inefficient representation of time steps could lead to the learning problem such as gradient explosion. Although in the paper the authors use hour as the time resolution, I doubt whether this can be generalized to all scenarios since it depends on the initial time points. If this is not generalizable, then the modelling of time dependency is less effcient than the time-independent but with extra context representing the state. Can the authors give more discussion on this point?
3. Can the authors give more details about the constructions of $g(\cdot)$ and $\beta\_{pt}$? In other words, why and how are the constructions yielded?
4. The authors lack many works on applying data-driven method to power system operations from IEEE transactions on smart grid and IEEE transactions on power systems. The comparisons with the related works in the same field is important. Can the authors give more related works in the discussion stage?


## References
[1] Ha, David, Andrew Dai, and Quoc V. Le. "Hypernetworks." arXiv preprint arXiv:1609.09106 (2016).


**Summary Of The Paper:**

This paper proposed using time-dependent parameters to model the value function for evaluating the cost-to-go function in power systems. This work can be regarded as an extension from the traditional adaptive dynamic programming (ADP) with the study on how to represent the value function that fits the uncertainty of the dynamics of power systems induced by the installation decentralized energy resources (DERs). This work shows the performance of the proposed method on a realistic simulation. The background knowledge is well written and clear.

**Summary Of The Review:**

In summary, this work is a good attempt on extending the traditional method with ML techniques. However, due to multiple concerns I have (see weaknesses), I recommend reject. On the other hand, if the authors can improve the work that can satisfy the standard of ICLR, the community should welcome such works dealing with the realistic problems.

---

> ### Author Response · Authors · 2021-11-23
> **Response to reviewer wkjh**
>
> Dear reviewer,
> We thank you for your time and dedication. Below we address the issues pointed out in the review:
> 1. Unless taken in a very wide sense, we do not believe that our work is a particular case of [1]. Sure, [1] is quite general as a "learnable model that drives other models". However, the very requirements of interpretability, parsimony and incorporation of expert information in a judicious way that we look for in our proposed method are lost if using such an approach.
> 1. Our proposed model does not involve recursion, nor is it a deep model where a gradient explosion (or vanishing) situation could occur. Regarding the state, variables describing the dynamic state of the system (inertia, time correlations etc.) are always present, no matter what architecture is used to represent the value function. Finally, we do not fully understand what the reviewer means by "generalization" with regard to time scales. As we mention in the revised manuscript, there are different time scales to be considered, each with its own dynamics, relevant state variables, and cost functions. The paper deals only with the short term model (STM), but the overall proposed method is directly applicable to all of them with no modifications.
> 1. We have now removed these details from the paper, as a proper description of such aspects is beyond the scope and focus of what we want to convey on the paper, and has shown to be a source of confusion.
> 1. Surely, there are hundreds of data-driven methods for power systems in the literature, but that includes a vast array of methods and problems of all sorts within the also vast and diverse field of Power Systems. We did not find ones that were close enough  so that they would be interesting to mention in this work. We welcome any specific citations that we may have missed and thank the reviewer in advance for them.

---

> > ### Comment · Reviewer_wkjh · 2021-11-23
> > **Re: Response to reviewer wkjh**
> >
> > Thanks for the authors' responses. I now give more explanations to the authors to address their confusions.
> >
> > 1.  I understand the authors' logic, as I am a half power guy. Nevertheless, the general logic in ML community is pursuing a general method, unless you can propose a method that specifies the existing general method with a huge improvement on some applications or extensive impact on a category of problems. I suggest the authors can improve the work as per the requirement.
> > 2.  "generalization" with regard to time scales means that whether your model can fit different time scales.
> > 3. I think removal of these uncertain points is not a good reaction. I suggest the authors can add them back in the later revision and give clearer explanations. In my view, these could be some kind of novelty that may attract me if these constructions can be well explained.
> > 4. Related work is not only for specifying how close your work is to the existing works. The most important effect is letting the guys in ML community know the whole landscape of the area that is unfamiliar to them and then give a fair evaluation, since you submit to a ML conference where few people understand power systems.

---

> > > ### Author Response · Authors · 2021-11-23
> > > **more clarification on the review**
> > >
> > > Dear reviewer, we appreciate your time and dedication,
> > >
> > > About 1) We agree that in some problems it may be a better representation to include time as one more state variable. But note that in this case, one more dimension is being added to the representation of the state space and with a discretization equal to the analyzed time horizon divided by the duration of the time step. In the proposed implementation, when proposing to train a model by time step, if Temporal Parsimony were not imposed (functions g () and the corresponding betas) we would be facing a similar situation. The grace of the proposed model is precisely to allow the imposition of said parsimony based on expert knowledge (such as trends and seasonality of the variables of interest) and on real operating restrictions (such as the impossibility of starting a combined cycle in 1 hour) .
> > >
> > > About 2) Regarding the passage of time, in principle, the algorithm does not present restrictions in that sense. Rather, the restrictions related to the passage of time are given by the quality of the System Model, the longer the energy balance between generation and demand is verified in larger time windows, filtering the details of maintaining the instantaneous balance of power. In the example shown in the work, the Short-Term Model of the Uruguayan system was selected with a horizon of 240 hours and hourly step. The same algorithm is applied to the entire chain of Long, Medium and Short Term models with horizons 5 years, 3 months and 10 days and with weekly, daily and hourly steps respectively. (It should be clarified that if the step is greater than the hour, the steps are subdivided into time bands calculated dynamically to reflect the power requirements at the hourly level).
> > > Regarding possible "gradient explosion" problems. Since there is a network for each time step and the gradient of each network is calculated independently, and we are not using recurrent networks, the gradient explosion problem does not occur. All the samples of the same time step are passed through the Back-Propagation algorithm of each network and its gradient with respect to the proper parameters is thus calculated. We have tested the proposed model with weekly, daily and horuly time-step with good results.
> > >
> > > About 3) It was not a reaction in the bad sense, we actually added appendix G clarifying the selection criteria of the g () functions and their penalties. Trying to clarify that it is in these criteria where part of the expert knowledge is incorporated.
> > >
> > > About 4) Thanks for the clarification and now we got the point perfectly. We could gladly add an appendix with an overview of the different ML application areas in the Powers System that would allow us to locate our work in that context to the ML community. But I think we can no longer modify the paper.

---

> > > > ### Comment · Reviewer_wkjh · 2021-11-25
> > > > **Re: more clarification on the review**
> > > >
> > > > Thanks for the authors' further responses.
> > > >
> > > > (1) First, I admit the empirical contribution to a specific application scenario of the paper, but ICLR is a top conference looking forward the general-purpose algorithm or theoretical contributions. For this reason, I suggest the author can consider submitting this work to KDD, a top conference for data-driven applications.
> > > >
> > > > (2) I understand the contents of the work. I suggest the authors think of 2 questions. Q1. How do you guarantee the network for time step t within the current episode can be generalised to any other episodes? Is there any insight or theoretical guarantee? Q2. Each time step with a model is not a acceptable solution from the view of ML community, since it would cause the application restricted by the algorithm, e.g., if the horizons extend, then the model complexity would be explosive. For this reason, I suggest the authors think about how to deal with this issue.
> > > >
> > > > (3) I agree with the authors' direction of the amendment. However, I suggest the authors can make more clarifications, not only by verbs. but also with some specific description on mathematical formulations. To my experience, these could improve the acceptance rate for the ML conference.
> > > >
> > > > (4) I suggest the authors add these in the next submission.

---

> > > > > ### Author Response · Authors · 2021-11-25
> > > > > **Re x 2: more clarification on the review**
> > > > >
> > > > > Dear reviewer,
> > > > > Thank you for your follow up. I'd like to make a few comments on your later ones.
> > > > > Regarding (1): we are aware of what ICLR is. The reason we submitted to this conference is that 1) it is an application, which is explicitly listed in the topics of interest of the CFP (quoted: "applications in audio, speech, robotics, neuroscience, computational biology, or any other field") and, second, because our proposed solution involves a novel formulation which  I honestly believe is relevant  to the ML community (of which I personally have been part for over 20 years).
> > > > > Regarding (2): Q1 perhaps this was not explained clearly enough in the paper: the episodes form a "sliding window". The time step at time t in the current step is the "warm start" model for time t-1 in the next episode.
> > > > > Q2: the key point in the paper, again perhaps not properly explained, is that the complexity of the ensemble of networks is controlled by the function g(), that is, the guiding function for the weights. In fact, this network is able to provide better results, with far less parameters, than a network trained over all the episode. Unfortunately, we did not have time to include this baseline model, which we only recently were able to properly construct. This is, from my point of view, the biggest omission in the current paper (which, notoriously, was not mentioned by any reviewer).
> > > > > (3) That is definitely true and also a weak point of this paper.  Again, we had no time to properly include many of the intermediate results (calibration, validation, convergence during training) that we have.
> > > > > (4) I believe the message was cut or something like that...

---

### Official Review · Reviewer_QHRg · 2021-10-31

**Correctness:** 2
**Technical Novelty And Significance:** 1
**Empirical Novelty And Significance:** 1
**Recommendation:** 1
**Confidence:** 4

**Details Of Ethics Concerns:**

No concerns.

**Main Review:**

I would say the only strength of this paper may be the application results yet it is difficult to be sure given that the method being compared to is not expected to perform well on a large and complex system. It may be currently in use as the authors are saying yet that should qualify their approach as a good option for whoever is interested in using it for the Uruguayan system but not as a scientific argument, in my opinion.

As for weaknesses, the paper seem to be a concatenation of well known and simple techniques with some specifics that are not justified. For example, why is the third order polynomial chosen in Section 4.4 and why are betas as penalties chosen as in Section 5. There are no numerical characterizations of the performance of the algorithm, no bounds on the errors, no proofs of convergence, etc. In general, the paper provides no technical justifications for the claims.


**Summary Of The Paper:**

The paper provides an approximation of the cost-to-go for a particular system. The particular system is the Uruguayan system. The results are compared to the approximate dynamic programming as provided in some classical books that are cited in the paper.

**Summary Of The Review:**

The paper does not provide technical justifications for the claims. The approach is compared to a classical technique that is not expected to be very efficient when applied to a large and complex system, in general.

---

> ### Author Response · Authors · 2021-11-23
> **Answer to Reviewer QHRg**
>
> We thank the reviewer for the comments. We believe that there are important points to clarify here:
> * **Regarding the comparison**: it is not clear from the comment whether, by "large and complex system", the reviewer refers to
>    a) the current system under study or b) a larger and more complex system.  In the first case, it should be pointed out that the classic Bellman Recursion algorithm is widely deployed to operate the power systems of most countries arould the world; variants such as SDDP offer some advantages, but are by no means a clear winner. In terms of *efficacy*, CBR provides excellent results. The main concern that drives this work is *efficiency*, as it requires large computational resources and does not scale well to *larger* problems. In the case of b), precisely, since CBR cannot be used, there is no way of performing a comparison and thus we must resort to a scenario were both methods can actually be run.
>
> * **Regarding the novelty**: our work, as most proposed works go, is a combination of existing methods, tools, techniques and recipes. The novelty in our work, as most works go, lies in how these methods are combined in order to produce an algorithm that is efficient and effective. For instance, the use of a parametric model for providing guidance and continuity to a sequence of similar (non-recurrent) networks has, as far as we know, not been published elsewhere (see comments to Reviewer wkhj).
> To achieve such efficacy and efficiency, it is key to exploit the prior expert information about the system.
> It is true that several specifics were not properly justified. We have now modified the manuscripts to address these problems so that the proposed method is better justified as a whole (see comments to Reviewer  dQUM)
>
> * **Use of polynomials**: we agree with the reviewer in that we did not provide enough justification for this critical choice. This derives from prior knowledge about the main state variables of an electric system, namely, the water level in reservoirs such as lakes. For the time period considered of 240 hours and a time step of 1 hour, the curves of these variables as a function of time are known to be well represented by Taylor expansions  of degree 3. As we expect these variables to drive the  fluctuations in the model parameters across time, we chose to experiment with polynomials of degrees from 1 to 4, confirming that no additional gains can be observed for orders above 3.
>
> * **Error bounds, convergence** In accordance with what the reviewer pointed to above, the proposed solution is a combination of existing and well understood methods whose theoretical properties, including error bounds and convergence, have been established for decades. Such are artificial neural networks (ANNs) as universal approximators, rate of convergence of Monte Carlo estimates, in particular using Common Random Numbers, or iterated value function approximation using parametric approximation methods such as ANNs.
>
> * **Numerical characterization** We do provide a thorough numerical characterization of the performance of the algorithm. This is not reflected in the main text. We have now added details on this in Appendix F.

---

### Official Review · Reviewer_dQUM · 2021-11-06

**Correctness:** 3
**Technical Novelty And Significance:** 1
**Empirical Novelty And Significance:** 4
**Recommendation:** 3
**Confidence:** 3

**Main Review:**

Strengths:
* The problem addressed in the paper is well-motivated from the perspective of both theory and practice, with direct application to a real-world system.
* The high-level approach and assumptions are clearly described.
* The performance of the method seems very competitive with respect to the existing baseline, though more details on the evaluation should be provided (see below).

Weaknesses/questions:
* In general, I think the paper lacks some concreteness on the exact power systems setting, which make certain details of the paper harder to grasp. For instance, how is the value function actually used during planning? Do the cost functions represent only power generation constraints, or also e.g. constraint violations?
* I was not able to understand why a different model would need to be fit to every time step, or why this would make sense in the power systems setting. Is the idea that T=24 hours and t=1 hour, and since load values follow a roughly daily fluctuation, similar networks are used at times of similar load? Again, without more concreteness on the specific setting being studied, this was hard to grasp.
* Some critical details were missing, which made it hard to grasp the quality or soundess of the underlying method. For instance:
  * Section 4.1: More details should be given on the common random numbers technique (either in the main body or in an appendix), as the way the ensembles/chronicles are generated presumably has large implications for performance.
  * Section 4.3: I am not fully sure how the "anchor values" were employed.
  * Section 4.4: Why does this particular choice of $g$ make sense in the power systems context?
* Section 4.2: In the update equation, should the terms with/without tildes be switched? I am not sure the current update as written mathematically follows clearly from the verbal description/I am not sure where all the terms are coming from.
* Isn't doing online forward simulation at each step potentially expensive?
* In general, it was hard to fully understand how the different steps presented in different sections of the paper fit together. An algorithm box would greatly enhance clarity.
* Additional details need to be provided in the experiments, in order to fully understand the efficacy of the proposed method. For instance:
  * Section 5: Some more details of the classic Bellman iteration-based method that is currently used could be provided. E.g., why is it currently so expensive?
  * Figure 3: More details on how the evaluation was conducted should be provided -- e.g. details of how $w_t$'s were generated in practice may have a large bearing on the results.
* Figure 4: It could be more thoroughly explained why parsimony between these two quantities is necessary/useful. Right now, the explanation given seems somewhat hand-wavy.

Minor points (not affecting my score):
* Page 2, Section 1.2: "planification" -> "planning"
* Section 4.2: Why was $\alpha = 0.3$ chosen?

**Summary Of The Paper:**

This paper provides a deep learning-based method to estimate value functions for multi-step dispatch on a power grid. This method entails training a different neural network to approximate the value function for each time step within a fixed horizon; supervision is provided via point-wise estimates of the states visited within rolled-out trajectories, as estimated via compound TD estimation. The method is demonstrated to have comparable performance to the "classic" Bellman iteration-based method that is currently used within the SimSEE dispatch system (currently employed by the Uruguayan electricity system operator), while running significantly (~150x) faster.

**Summary Of The Review:**

The problem studied in this paper is important and well-motivated, and at a high level, the proposed approach seems reasonable and to work well. However, descriptions of several critical components of the theory and evaluation are either vague or not provided, making it difficult to truly evaluate the soundness and performance of the method.

---

> ### Author Response · Authors · 2021-11-23
> **Answer to reviewer dQUM**
>
>
> We thank the reviewer for the thorough and careful review, as well as the constructive comments given. Overall, we agree that important contextual information was left out which makes it difficult to understand the many technical decisions involved. Below we provide some answers and corrections based on them:
> * **Meaning of the value function**: this is actually a complex issue, as planning for different time horizons involves different decisions (from instantaneously opening and closing gates to purchasing fuel shipments with months in advance), and thus different value functions to guide such decisions. We have included a brief description on how the value function is constructed in for different time scales in Appendix B.
> * **Constraints and their violations**: we now provide some basic details on this subject in Appendix C.
> We hope that these additions help in understanding the overall problem.
> * **Training at each time step**: It is also true that not enough justification was provided for the need to re-train the system at each time slot. The short answer is that the system itself is continually changing. Even if we model for a given horizon, once a concrete action has been taken and new input  information is available (demants, forecasts, etc.), we need to re-compute the policies. A slightly longer description is now provided in Appendix D.
> * **Common Random Numbers**  This technique is conceptually very simple and amounts to what is briefly described in Section 4.1, that is, to fix a number of pseudo-random number sequences for simulating system inputs; this is a trade-off which reduces the variance of the Monte Carlo estimations of the value function at the cost of an increaed bias. For more details, we believe that the reader should directly refer to the citation provided, rather than providing a more technical introduction in an appendix.
> * **Anchor values** We have now omitted the details regarding to "model chaining". Although this is an important aspect of the operation of the real system, it deviates from the focus of the paper, and requires a significant amount of additional information in order to be properly understood.
> * **Update equation in Sec. 4.2** Indeed this was a mistake. We thank the reviewer for pointing it out. This has now been corrected.
> * **Online update** Actually, there is no *online* adaptation of the algorithm. At the smallest scale, we have a full hour at our disposal to recompute the optimal policy in an off-line fashion. We have now added a diagram for the whole system, with its updates, in Appendix A, which we hope will clarify this matter.
> * **Overall algorithm** We have now added a diagram in Appendix A which we hope will serve to describe the algorithm as a whole.
> * **Computational cost of Bellman Recursion** We have now added Appendix E with an explicit calculation of the number of value function estimations required in this case.
> * **Evaluation of the computed policies** Indeed, we omitted details in this respect due to space restrictions. We have now added more details on this in Appendix F
> * **The need for parsimony** We mistakenly omitted this important piece of information in the drive for meeting the page limit;  this is by no means an arbitrary decision. We provide a more detailed justification on this matter in Appendix G.
> * **Choice of learning rate** The value of $0.3$ was observed to produce the best overall results in our simulations. This "learning rate", however, is not to be confused with the similar term used to describe step sizes in descent algorithms, but rather an update factor to integrate past learned information in subsequent iterations.

---

> > ### Comment · Reviewer_dQUM · 2021-12-01
> > **Response to authors**
> >
> > Thanks to the authors for their careful consideration of my points, and for the resulting clarifications.
> >
> > ### Overarching comment on clarity
> >
> > As an overarching comment, while I appreciate the additional details given in the appendix, I think some of these additions and details would need to be more tightly integrated into the main body of the paper in order for the main body to stand alone. This often does not take many sentences -- e.g., I think the extended explanation in Appendix B for the meaning of the value function is helpful, but in the main body of the paper, I am missing the quick summary of e.g. "At a high level, the value function captures the amount of power generation/power generation capacity available as well as associated demand. Such value functions are used to trigger operational decisions such as fuel purchases (when used for planning a few months ahead) or grid balancing operations (in the real-time setting)."
> >
> > In general, I think the paper does need to be better organized for these details to come through. Right now, the paper jumps between high level power system motivation, generic background on ADP, back to power system background, back to pure algorithmic discussion, etc. - but these details are not really woven together. Even for someone like myself with a background in both ML and power systems, this makes the overall thread of the paper hard to follow.
> >
> > ### Responses to individual points
> >
> > > Meaning of the value function: this is actually a complex issue, as planning for different time horizons involves different decisions (from instantaneously opening and closing gates to purchasing fuel shipments with months in advance), and thus different value functions to guide such decisions. We have included a brief description on how the value function is constructed in for different time scales in Appendix B.
> >
> > Thank you for the details. As mentioned above, I think it could be good to hint at the intuition behind this more strongly in the main body.
> >
> > > Constraints and their violations: we now provide some basic details on this subject in Appendix C. We hope that these additions help in understanding the overall problem.
> >
> > To me, this explanation still seems a little vague.
> >
> > > Training at each time step: It is also true that not enough justification was provided for the need to re-train the system at each time slot. The short answer is that the system itself is continually changing. Even if we model for a given horizon, once a concrete action has been taken and new input information is available (demants, forecasts, etc.), we need to re-compute the policies. A slightly longer description is now provided in Appendix D.
> >
> > My question was not actually about this - I fully understand why the model needs to be retrained. But, I would have liked to see a little bit of intuition for how $T$ and $t$ were chosen, and why a sliding window strategy makes sense (i.e., some concrete argument about the periodicity of the underlying system).
> >
> > > Common Random Numbers This technique is conceptually very simple and amounts to what is briefly described in Section 4.1, that is, to fix a number of pseudo-random number sequences for simulating system inputs; this is a trade-off which reduces the variance of the Monte Carlo estimations of the value function at the cost of an increaed bias. For more details, we believe that the reader should directly refer to the citation provided, rather than providing a more technical introduction in an appendix.
> >
> > Absolutely, that makes sense. That said, I still think there could be a half-sentence explanation on how the diversity of lack of diversity of the specific samples may or may not impact performance in this case.
> >
> > > Anchor values We have now omitted the details regarding to "model chaining". Although this is an important aspect of the operation of the real system, it deviates from the focus of the paper, and requires a significant amount of additional information in order to be properly understood.
> >
> > I think it would have been good to keep this in, but maybe move it to an appendix.
> >
> > > Update equation in Sec. 4.2 Indeed this was a mistake. We thank the reviewer for pointing it out. This has now been corrected.
> >
> > Fantastic, thanks!
> >
> > > Online update Actually, there is no online adaptation of the algorithm. At the smallest scale, we have a full hour at our disposal to recompute the optimal policy in an off-line fashion. We have now added a diagram for the whole system, with its updates, in Appendix A, which we hope will clarify this matter.
> >
> > Got it, thank you for the clarification.

---

> > > ### Comment · Reviewer_dQUM · 2021-12-01
> > > **Response to authors, part 2**
> > >
> > > > Overall algorithm We have now added a diagram in Appendix A which we hope will serve to describe the algorithm as a whole.
> > >
> > > Thank you for adding this. I do think this could be made more concrete, however (e.g., references to specific sections or equations, and avoidance of jargon like "chronicles" but instead spelling out what is actually happening).
> > >
> > > > Computational cost of Bellman Recursion We have now added Appendix E with an explicit calculation of the number of value function estimations required in this case.
> > >
> > > Thanks for adding this. Is Classic Bellman’s Recursion indeed what is being used currently? Or some variant? That is the detail I was missing, and that I think should be made clearer.
> > >
> > > > Evaluation of the computed policies Indeed, we omitted details in this respect due to space restrictions. We have now added more details on this in Appendix F
> > >
> > > Thank you, this is helpful.
> > >
> > > > The need for parsimony We mistakenly omitted this important piece of information in the drive for meeting the page limit; this is by no means an arbitrary decision. We provide a more detailed justification on this matter in Appendix G.
> > >
> > > This is helpful. However, more concreteness is needed on how "operating instructions arise directly from the derivatives of the
> > > value function" for this to fully make sense.
> > >
> > > > Choice of learning rate The value of  was observed to produce the best overall results in our simulations. This "learning rate", however, is not to be confused with the similar term used to describe step sizes in descent algorithms, but rather an update factor to integrate past learned information in subsequent iterations.
> > >
> > > Thanks for clarifying.

---

### Decision · Program_Chairs · 2022-01-20

**Decision:**

Reject

**Comment:**

The reviews are of good quality. The responses by the authors are commendable, but ICLR is selective and reviewers continue to feel that important choices in the research are not sufficiently clear and fully justified.